# Evaluation of dynamic cerebrovascular autoregulation during liver transplantation

**Bente Marei Wolpert[1], David Jonas Rothgerber[1], Ann Kristin Rosner [1], Malte Brunier[1], Robert Kuchen[2], Patrick Schramm [3☯, Eva-Verena Griemert [1☯ \***

1 Department of Anesthesiology, University Medical Centre of the Johannes-Gutenberg University Mainz, Mainz, Germany, 2 Institute of Medical Biostatistics, Epidemiology and Informatics (IMBEI), University Medical Centre of the Johannes-Gutenberg University Mainz, Mainz, Germany, 3 Department of Neurology, University Hospital of the Justus-Liebig-University Giessen, Giessen, Germany

☯ These authors contributed equally to this work.
\* ev.griemert@uni-mainz.de

**Data Availability Statement:** All relevant data are within the paper and its Supporting Information files.

## Abstract

### Background

Cerebrovascular autoregulation in patients with acute and chronic liver failure is often impaired, yet an intact autoregulation is essential for the demand-driven supply of oxygenated blood to the brain. It is unclear, whether there is a connection between cerebrovascular autoregulation during liver transplantation (LTX) and the underlying disease, and if perioperative anesthesiologic consequences can result from this.

### Methods

In this prospective observational pilot study, data of twenty patients (35% female) undergoing LTX were analyzed. Cerebral blood velocity was measured using transcranial doppler sonography and was correlated with arterial blood pressure. The integrity of dynamic cerebrovascular autoregulation (dCA) was evaluated in the frequency domain through transfer function analysis (TFA). Standard clinical parameters were recorded. Mixed one-way ANOVA and generalized estimating equations were fitted to data involving repeated measurements on the same patient. For all other correlation analyses, Spearman's rank correlation coefficient (Spearman's-Rho) was used.

### Results

Indications of impaired dCA are seen in frequency domain during different phases of LTX. No correlation was found between various parameter of dCA and primary disease, delirium, laboratory values, length of ICU or hospital stay, mortality or surgical technique.

### Conclusions

Although in most cases the dCA has been impaired during LTX, the heterogeneity of the underlying diseases seems to be too diverse to draw valid conclusions from this observational pilot study.

**Funding:** The authors received no specific funding for this work.

**Competing interests:** The authors have declared that no competing interests exist.

**Abbreviations:** ABP, arterial blood pressure; ASA, American Society of Anesthesiologists; dCA, dynamic cerebrovascular autoregulation; CAM-ICU, Confusion Assessment Method for the Intensive Care Unit; CBF, cerebral blood flow; CBv, cerebral blood velocity; CI, cardiac index; CO, cardiac output; $CO_2$, carbon dioxide; CPP, cerebral perfusion pressure; HE, hepatic encephalopathy; HR, heart rate; ICP, intracranial pressure; ICU, intensive care unit; INR, international normalized ratio; LTX, liver transplantation; MAC, minimum alveolar anesthetic concentration; MAP, mean arterial pressure; MCBv, mean cerebral blood velocity; MELD, model for the end stage of liver disease; Mxa, autoregulation index; $PaCO_2$, arterial carbon dioxide partial pressure; $rSO_2$, regional oxygen saturation; TCD, transcranial doppler sonography.

## Introduction

Dynamic cerebrovascular autoregulation (dCA) is often impaired in patients with acute and chronic liver failure [1, 2]. An intact dCA is essential for the demand-driven supply of oxygenated blood to the brain. In healthy individuals dCA ensures, that cerebral blood flow (CBF) is kept relatively constant independent of mean arterial blood pressure (MAP) and carbon dioxide partial pressure ($p_aCO_2$) as long as MAP is within the lower and upper limits of dCA, i.e. 60 to 150 mmHg [3]. If dCA is disrupted, blood flow to the brain becomes pressure-passive, which can cause both ischemic and hyperemic neurologic disorders [4, 5]. Patients with terminal liver disease often exhibit a high cardiac output (CO) combined with low vascular resistance, making them particularly susceptible to sudden drops in MAP [6, 7]. In addition, due to the surgery, severe blood loss and fluid shifts as well as an imbalance in homeostasis may occur and aggravate the situation. This may result in a decrease in cerebral perfusion pressure (CPP) with consecutive lowering of cerebral blood flow (CBF) and, therefore, risk of cerebral hypoxia during orthotopic liver transplantation (LTX) [5]. On the other hand, an increase in MAP [8] or $p_aCO_2$ [9] in patients with an acute or chronic liver failure accompanied by an impaired dCA leads to a marked increase in cerebral perfusion with the risk of developing cerebral oedema and can entail an uncontrolled increase in intracranial pressure (ICP) [10]. During LTX, patients are particularly at risk when, depending on the surgical approach, the vena cava may be cross clamped. Reperfusion of the donor organ is also a critical phase, during which hemodynamic instability is common, and cerebral blood flow may vary. However, the cause of the alteration in CBF and impaired dCA is not fully understood but correlates with liver function [1]. Moreover, even subclinical encephalopathy, often associated with terminal liver disease, increases the risk for adverse outcome and mortality after LTX [11]. In this observational pilot study, dCA was measured using transcranial doppler sonography (TCD) in patients undergoing LTX, focusing on any link between the level as well as changes of dCA and primary disease, pre-existing hepatic encephalopathy (HE) or renal failure with dialysis, laboratory parameters, and surgical technique.

## Methods

The study was approved by the local ethics committee of Rhineland-Palatinate, Germany (approval number: 837.041.10 (7050)) and registered at ClinicalTrials.gov (NCT01597102). As this was an observational cohort pilot study, the STROBE protocol was applied [12].

From July 1, 2020, until December 2, 2021, patients listed for LTX at the University Medical Centre Mainz, Germany, were included, and written informed consent was obtained. Exclusion criteria were as follows: age less than 18 years, pregnancy, and a missing temporal window for TCD.

### Anesthetic management

All patients received general anesthesia. Therefore, 0.2–0.5 µg $kg^{-1}$ sufentanil (Sufentanil-hameln, Hameln Pharma GmbH, Hameln, Germany) was administered intravenously, followed by 1–2 mg$kg^{-1}$ propofol (Propofol-®Lipuro, B. Braun Melsungen AG, Melsungen, Germany) and 0.9 mg$kg^{-1}$ rocuronium (Rocuronium Inresa, Inresa Arzneimittel GmbH, Freiburg, Germany).

Anesthesia was maintained with sevoflurane (Sevorane®, AbbVie Inc. North Chicago, Illinois, USA), at an age-adjusted minimum alveolar anesthetic concentration (MAC) of 0.6 to 1.0 calculated by the monitoring software (Pallas, Dräger, Lübeck, Germany), and with repeated injections of sufentanil and atracurium if necessary. Ventilation was adapted to maintain an arterial partial pressure of carbon dioxide ($p_aCO_2$) between 35 and 45 mmHg. MAP

was maintained between 60 and 90 mmHg based on institutional standard operating procedures.

The patient was placed in the supine position. The head was positioned in a foam pillow ensuring that it was straight in order to guarantee venous drainage and prevent compression of the neck.

Regional cerebral oxygen saturation ($rSO_2$) was assessed continuously via the INVOS™ cerebral oximeter (Medtronic, MN, USA). The sensors of the oximeter were positioned on the right and left forehead in the frontotemporal position before induction of anesthesia.

## dCA measurement

The integrity of the dCA was assessed following the induction of anesthesia using simultaneous recording of cerebral blood velocity (CBv) and invasively measured arterial blood pressure (ABP). CBv was continuously measured using TCD in the left middle cerebral artery, with an insonation depth of 40–60 mm. Therefore the 2-MHz Doppler probe (Doppler Box©; DWL, Sipplingen, Germany) was positioned in the temporal ultrasound window above the zygomatic arch and fixed with the help of a helmet.

Invasive arterial blood pressure was measured in the radial artery (M10006B/MP70, Philips Medical Systems, Eindhoven, Netherlands). The analogue curves of ABP and CBv were synchronously recorded and digitized at a sampling rate of 100 Hz using the Doppler Box©. Prior to dCA analysis, the recorded curves underwent visual inspection, and any artifacts were removed from the recordings. The integrity of dCA was evaluated in the frequency domain through transfer function analysis (TFA), which involved determining coherence, gain, and phase shift [13–15].

The two pertinent parameters for assessing the integrity of dCA are the gain and the phase shift. These parameters were computed following the guidelines for three frequency ranges: the very low-frequency band (0.02–0.07 Hz), the low-frequency band (0.07–0.2 Hz), and the high-frequency band (0.2–0.5 Hz). Gain and phase were considered for further evaluation only if the coherence threshold was satisfactorily [13, 16, 17]. A lower gain and a higher phase indicate intact dCA in the various frequency bands.

Additional physiological data (MAP, CI, $p_aCO_2$, MAC and norepinephrine rates) were recorded every 5 or 15 minutes, respectively. For the analysis, an average value was calculated for each phase and patient.

Before LTX, blood samples were collected from all patients. Total bilirubin, Quick value, and levels of aspartate aminotransferase, alanine aminotransferase, and glutamate dehydrogenase were among the parameters assessed. Furthermore, the Model for End Stage Liver Disease (MELD) score was determined. The score is used for the classification of the severity of liver disease and the prioritization of patients on the waiting list for liver transplantation. The complex calculation of the MELD score is based on serum bilirubin, serum creatinine, and the international normalized ratio (INR) of prothrombin time [18]. The maximum achievable score is 40. These preoperatively obtained blood samples were utilized for the statistical analysis.

## Intensive care unit stay

After surgery, patients were admitted to the intensive care unit, preferably already extubated. Standardized examinations using the CAM-ICU score for delirium were conducted once per shift, i.e., three times a day.

## Statistical analysis

The primary objective was to investigate changes in the integrity of dCA during the different phases of LTX. Secondary outcomes included examining the potential influence of pre-operative disorders and the type of liver failure on dCA, as well as assessing delirium within the first 4 postoperative days, the length of stay at ICU, hospital stay after LTX and, mortality within the first year after LTX.

The estimated case number by SAS-ANOVA yielded a patient number of 20 with a power of > 0.9 and a first kind error of 0.05.

To test the null hypothesis that the mean dCA values are identical across the three different phases of LTX (overall test), mixed one-way ANOVAs with random intercepts for each patient were conducted. Since the mixed-model algorithms failed to converge in the very-low-frequency domain, generalized estimating equations were used as an alternative. When the overall tests were significant, post-hoc tests with pairwise comparisons were performed to assess the null hypothesis that the mean dCA values of the respective LTX phases are identical. A Sidak adjustment was applied to control for multiple comparisons. In the statistical analyses of the secondary outcomes, several important parameters were tested for significant correlations with dCA parameters using Spearman's rho.

All statistical analyses were performed using SPSS (version 29, SPSS Inc., Chicago, IL, USA and RStudio, version 2023.09.1, Posit Software, USA; R version 4.3.1, R Foundation, Austria). For all analyses p-values below $\alpha = 0.05$ were considered significant.

## Results

### Population

In total, 24 patients undergoing LTX were initially included, with data from 20 patients being available for analysis (7 females and 13 males). Two patients were excluded due to a missing temporal ultrasound window. Additionally, one patient was excluded because of an intraoperative termination of the surgery, and another patient due to a technical error during data transmission.

The mean age was 52 ± 13 years. Thirteen patients had a score of 3 according to the physical status classification of the American Society of Anesthesiologists (ASA), 4 patients were ASA 2, and 3 were ASA 4 (Table 1 and S1 Table).

### Physiological data

The physiological data in relation to the phase of liver transplantation (LTX) are presented in Table 2.

MAP in the preparation phase was 11 (95%-CI [2, 21]) mmHg and in the anhepatic phase 18 (95%-CI [9, 28]) mmHg higher than in the reperfusion phase (F = 11.12, p < 0.001). MCBv in the reperfusion phase was about 10 (95%-CI [0.2, 19]) cms$^{-2}$ higher as in the anhepatic phase (F = 3.81, p = 0.032). The cardiac index (CI) in the anhepatic phase was 1.5 (95%-CI [0.5, 2.5]) lmin$^{-1}$m$^{-2}$ higher than in the reperfusion phase (F = 7.46, p = 0.002). The heart rate (HR) in the preparation phase was on average 14 (95%-CI [5, 24]) min$^{-1}$ higher than in the anhepatic phase (F = 6.96, p = 0.003). The arterial $CO_2$ partial pressure ($p_aCO_2$) in the anhepatic phase was 4.1 (95%-CI [1.0, 7.3]) mmHg and in the reperfusion phase 5.6 (95%-CI [2.5, 8.8]) mmHg higher than in the preparation phase (F = 10.76, p < 0.001). The end-expiratory sevoflurane concentration (F = 3.33, p = 0.05) and the need for norepinephrine (F = 3.32, p = 0.054) did not differ between the phases of LTX.

**Table 1. Patient population.**

| | |
|---|---|
| **ASA (mean)** | 2,95 |
| **Sex** | |
| female (in %) | 35 |
| male (in %) | 65 |
| **Age (yrs.; mean + STD)** | 52 ± 13 |
| **MELD Score (median; [interquartile range])** | 25,5 [17,75; 30] |
| **Dialysis (in %)** | 20 |
| **HE (in %)** | 30 |
| **Onset of liver failure** | |
| ACLF (in %) | 25 |
| ALF (in %) | 5 |
| CLF (in %) | 70 |
| **Delirium within 4 days (at ICU) (in %)** | 20 |
| **One-year mortality (in %)** | 15 |

ACLF: acute on chronic liver failure; ALF: acute liver failure; ASA: American Society of Anesthesiologists physiology score; CLF: chronic liver failure; HE: hepatic encephalopathy; ICU: Intensive care unit; MELD: model for the end stage of liver disease; STD: standard deviation; yrs.: years

## Primary outcome

The primary outcome was defined as difference of the integrity of dCA between the phases of LTX (Table 3).

Calculations in the frequency domain were conducted using the parameters gain and phase. This analysis was performed across three predefined frequency domains: very low, low, and high frequency. In the very low frequency domain, the gain was lower in the preparation phase compared to the anhepatic ($p = 0.004$) and reperfusion phase ($p = 0.006$) (Table 4 and Fig 1).

Meanwhile, there were no significant differences in phase among the phases of LTX ($p = 0.911$; Fig 2).

An overall test revealed significant differences between the mean gain in the low frequency domain ($p = 0.038$). Conducting post-hoc pairwise comparisons, only the mean gain between the preparation and the reperfusion phase differed significantly ($p = 0.046$, Fig 3).

**Table 2. Physiological data in dependence of the phase of LTX.**

| | Phase of LTX | | | significance |
|---|---|---|---|---|
| | Preparation | Anhepatic | Reperfusion | |
| **MAP [mmHg]** | 71 ± 12 | 77 ± 11 | 59 ± 16 | < 0.001 |
| MCBv [cms$^{-2}$] | 49 ± 22 | 47 ± 15 | 57 ± 21 | 0.032 |
| CI [lmin$^{-1}$m$^{-2}$] | 4.4 ± 1.9 | 5.1 ± 1.8 | 3.5 ± 1.2 | 0.002 |
| HR [min$^{-1}$] | 91 ± 21 | 77 ± 15 | 58 ± 15 | 0.003 |
| p$_a$CO$_2$ [mmHg] | 36 ± 5 | 40 ± 6 | 42 ± 6 | < 0.001 |
| **expSevo [MAC]** | 0.95 ± 0.36 | 0.85 ± 0.25 | 0.89 ± 0.32 | 0.05 |
| Norepinephrine [µgkg$^{-1}$min$^{-1}$] | 0.18 [0.10, 0.30] | 0.16 [0.07, 2.29] | 0.23 [0.12, 0.38] | 0.054 |

CI: cardiac index; p$_a$CO$_2$: arterial CO$_2$ partial pressure; expSevo: end-expiratory sevoflurane concentration; HR: heart rate; MAC: minimal alveolar concentration; MAP: mean arterial pressure; MCBv: mean cerebral blood velocity, LTX: liver transplantation. Statistical analysis was performed using one-way mixed-effects ANOVA. Data are shown as mean ± standard deviation or median [interquartile range].

**Table 3. Parameters of the dCA in dependence of the phase of LTX.**

|  | Preparation, N = 20 | Anhepatic, N = 20 | Reperfusion, N = 20 | P-value |
|---|---|---|---|---|
| **Gain, Very Low Freq.** |  |  |  | < 0.001 |
| Mean ± SD | 0.18 ± 0.27 | 1.01 ± 1.15 | 0.80 ± 0.93 |  |
| Median (Range) | 0.07 (0.01–1.19) | 0.29 (0.02–3.50) | 0.30 (0.01–3.19) |  |
| NA | 0 | 1 | 0 |  |
| **Phase, Very Low Freq.** |  |  |  | 0.911 |
| Mean ± SD | 60.68 ± 51.27 | 57.23 ± 41.16 | 53.76 (41.79 |  |
| Median (Range) | 57.50 (0.84–158.50) | 49.79 (4.11–135.55) | 46.02 (6.88–157.33) |  |
| NA | 5 | 5 | 6 |  |
| **Gain, Low Freq.** |  |  |  | 0.038 |
| Mean ± SD | 0.19 ± 0.25 | 1.02 ± 1.06 | 1.22 ± 1.96 |  |
| Median (Range) | 0.11 (0.01–0.83) | 0.63 (0.01–3.04) | 0.74 (0.02–7.85) |  |
| NA | 0 | 1 | 0 |  |
| **Phase, Low Freq.** |  |  |  | 0.935 |
| Mean ± SD | 7.98 ± 26.44 | 8.40 ± 31.81 | 5.36 ± 31.53 |  |
| Median (Range) | 1.57 (-22.35–73.05) | -0.40 (-33.10–78.84) | -3.36 (-48.60–87.29) |  |
| NA | 0 | 1 | 0 |  |
| **Gain, High Freq.** |  |  |  | 0.059 |
| Mean ± SD | 0.28 ± 0.41 | 0.8 ± 0.86 | 1.08 ± 1.64 |  |
| Median (Range) | 0.13 (0.01–1.45) | 0.43 (0.02–2.79) | 0.73 (0.07–7.58) |  |
| NA | 0 | 1 | 0 |  |
| **Phase, High Freq.** |  |  |  | 0.722 |
| Mean ± SD | -2.76 ± 33.57 | 1.90 ± 18.47 | 1.85 ± 19.55 |  |
| Median (Range) | -6.75 (-82.25–86.37) | 0.71 (-31.62–48.98) | -2.39 (-27.80–45.39) |  |
| NA | 0 | 1 | 0 |  |

LTX: liver transplantation, Freq.: frequency; NA: not available; very low frequency domain (0.02–0.07 Hz), the low frequency domain (0.07–0.2 Hz) and the high frequency domain (0.2–0.5 Hz). Statistical analyses were performed using one-way mixed-effects ANOVA for the high and low frequency domains and generalized estimating equations for the very low frequency domain.

However, there were no significant differences in phase of the low frequency domain analysis among the phases of LTX (p = 0.935; Fig 4).

In the high frequency domain, neither the gain (p = 0.059 nor the phase (p = 0.722) showed differences among the phases of the LTX.

## Secondary outcomes

Further analysis of influencing factors for dCA was conducted using gain, and phase in the low frequency band. No associations were found between the underlying disease, and the integrity of dCA.

Regarding the postoperative outcome parameters, 20% of patients developed delirium within the first 4 days at ICU after LTX. The incidence of a delirium was associated with a higher phase (38 vs. -4, p < 0.01) during the reperfusion period, while the other parameters did not differ during all time points.

The mean duration at ICU after LTX was 10 days (range from 1 to 62 days). The mean duration in hospital until discharge was 31 days (8 to 94 days). There was no correlation with the various parameter of dCA and the length of stay at ICU or hospital.

**Table 4. P-values of pairwise comparisons of the different phases of LTX.**

| Frequency/Comparison | Preparation–Anhepatic | Preparation–Reperfusion | Anhepatic–Reperfusion |
|---|---|---|---|
| Gain, Very Low Frequency | 0.004 | 0.006 | 0.899 |
| Gain, Low Frequency | 0.151 | 0.046 | 0.945 |
| Gain, High Frequency | 0.336 | 0.058 | 0.790 |

LTX: Liver Transplantation, P-values of pairwise comparisons when applying a Sidak correction for multiple comparisons to the models estimated in Table 3.

Within one year, 3 of 20 patients after LTX died (15%). There was no association detectable between the integrity of dCA and one-year mortality.

On average, patients received 5.7 red blood cell units, 8.3 fresh frozen plasma and 0.65 platelet concentrates during transplantation. Here, too, there was no correlation with the parameter of dCA. Moreover, no association was found between parameters of dCA and age,

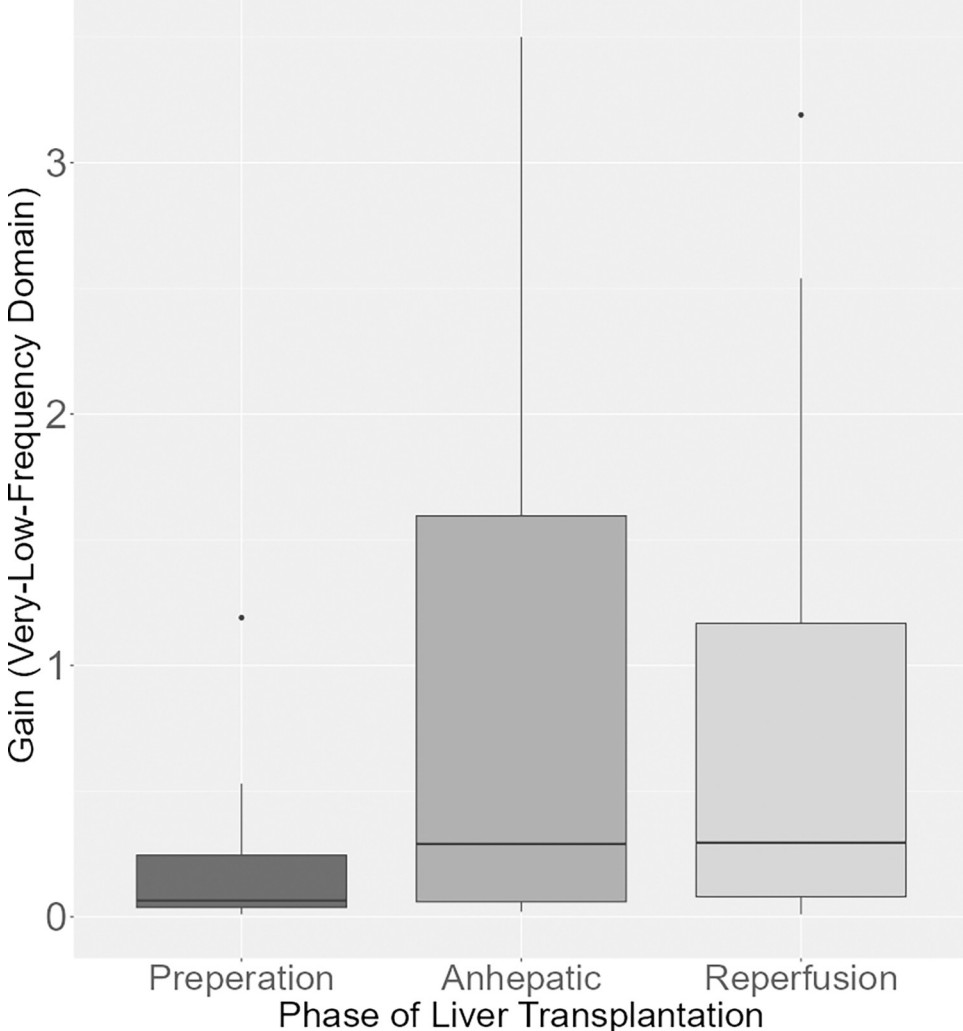

**Fig 1. Gain (very low frequency domain) during the phases of LTX.** Boxplot of the gain in the very low frequency domain. In the very low frequency domain, the gain was lower in the preparation phase compared to the anhepatic (p = 0.004) and reperfusion phase (p = 0.006).

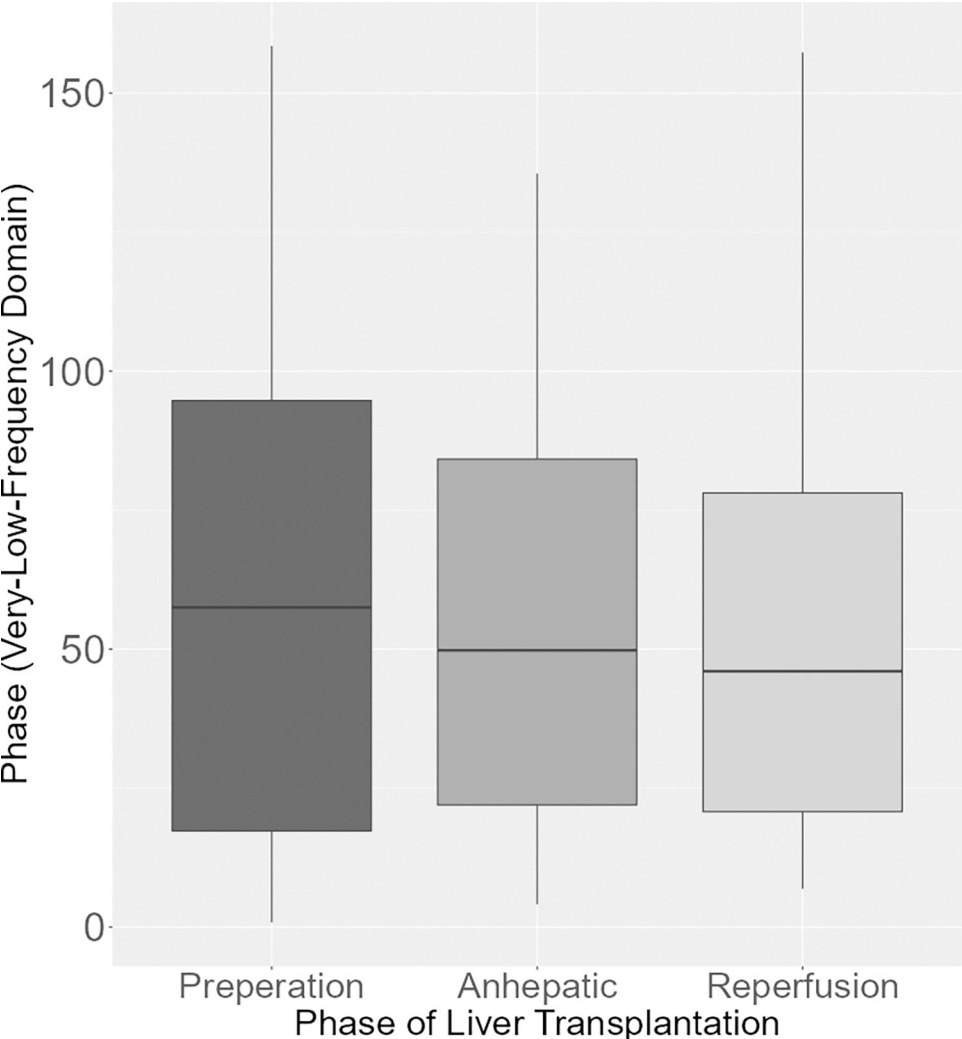

**Fig 2. Phase (very low frequency domain) during the phases of LTX.** Boxplot of the phase in the very low frequency domain showing no differences during the preparation, anhepatic and reperfusion phase during LTX.

MELD score, bilirubin levels, Quick, the level of aspartate aminotransferase, alanine aminotransferase and glutamate dehydrogenase or preoperative dialysis requirement (20%).

Taking a closer look at the surgical technique used, no correlation with parameters of dCA can be established here either. 35% of the patients received a total vena cava replacement, whereas 65% were operated on using the piggyback technique. Also, it does not seem to influence dCA whether the newly transplanted organ is flushed with blood or crystalloids during reperfusion.

## Discussion

The primary outcome was defined as difference of the integrity of dCA between the phases of LTX. For a considerable time, dCA was often evaluated in the time domain through the calculation of the Mxa (autoregulation index) as an indicator of its integrity [19]. In one of their latest papers, Olsen et al. effectively highlighted the challenges Mxa encounters in terms of methodological consistency, validity, and reliability and why it should no longer be used [20].

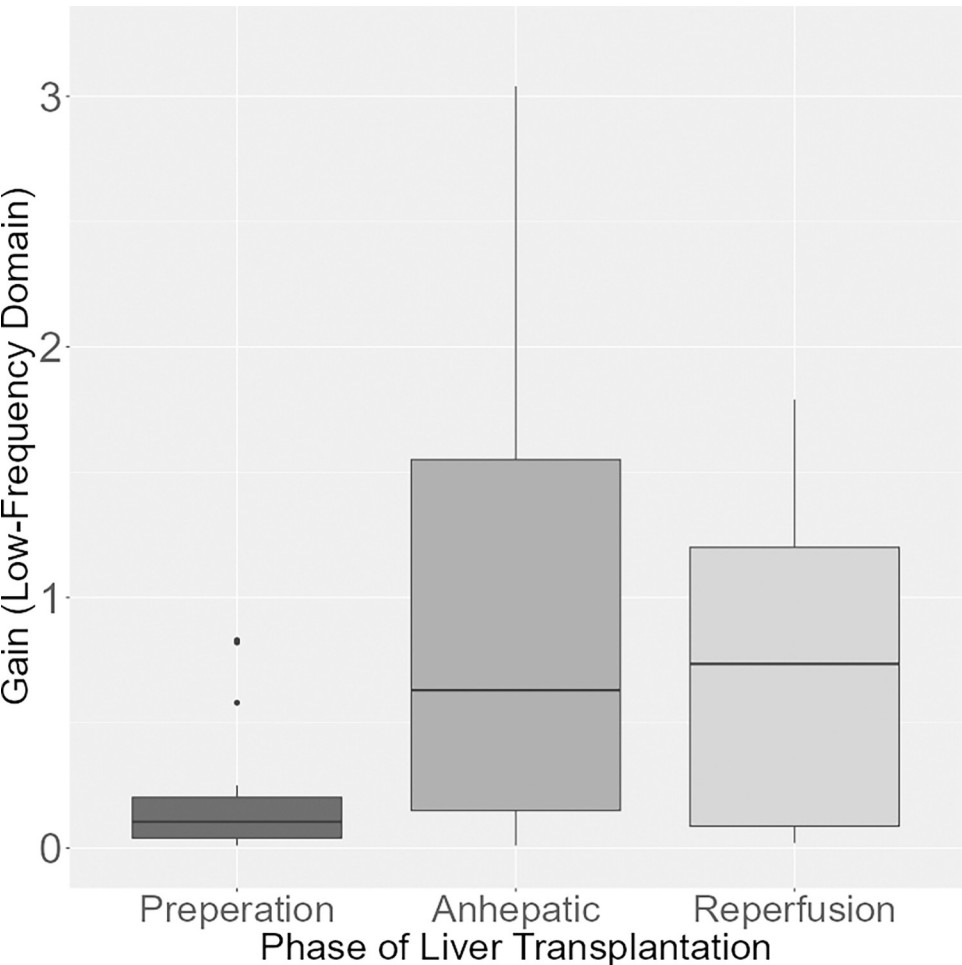

**Fig 3. Gain (low frequency domain) during the phases of LTX.** Boxplot of the gain in the low frequency domain. The gain was lower in the preparation phase compared to the reperfusion phase (p = 0.046).

Additionally, dCA is more of a continuous measure rather than a binary one. Therefore, establishing a precise cutoff value, such as the 0.3 threshold used by Sorrentino et al. in the context of head injury [21], is challenging in general. Therefore, the frequency domain was analyzed, and a lower gain and higher phase suggest normally an intact dCA in the various frequency bands. Our results exhibited an impaired dCA in the reperfusion phase compared to the preparation phase in the very low and low frequency band regarding gain. No differences were observed in the frequency domain analysis of phase. In summary, the results of this observational pilot study provide a subtle indication of a more intact dCA during the preparation phase.

## Population

Possibly, the heterogeneity of the underlying diseases (S1 Table) was too diverse, and the number of investigated patients too small to identify correlations between disturbed dCA and the specified outcome parameters. Two similarly structured studies exist. In one study, six patients with acute liver failure were examined revealing impaired autoregulation detected prior to transplantation [1]. In the other study, nine patients with chronic liver failure were

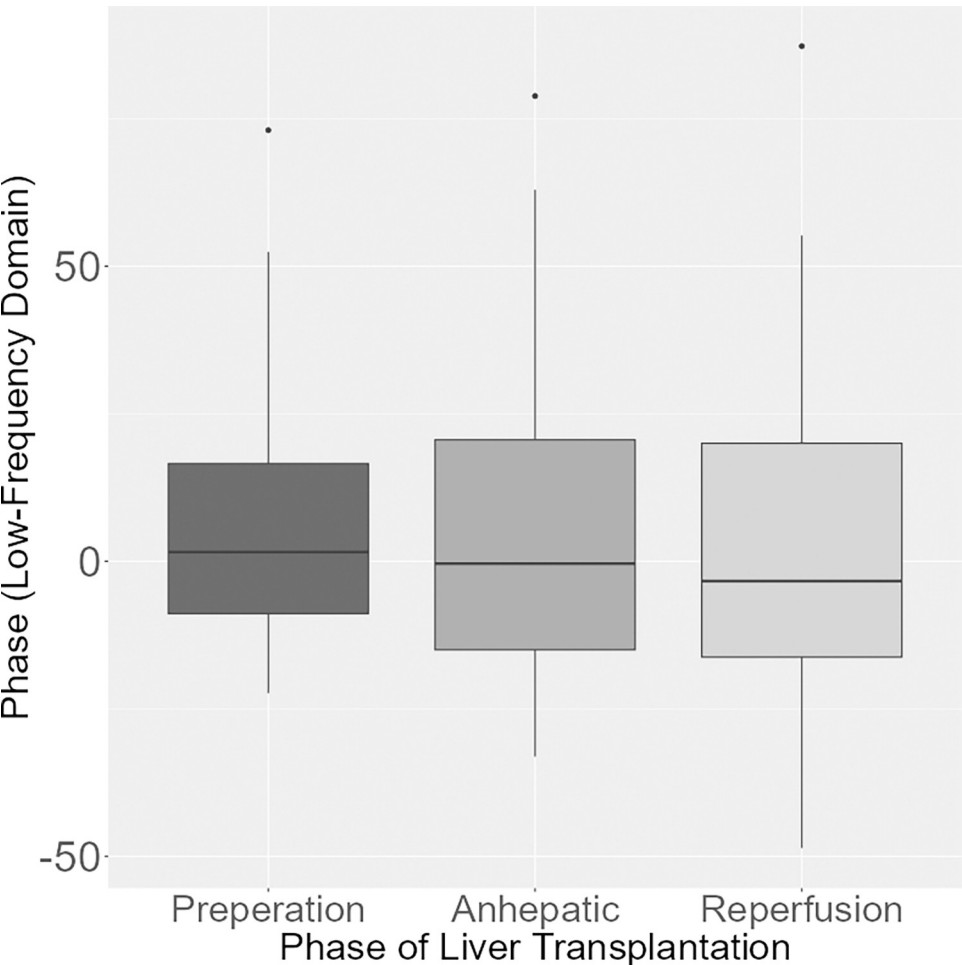

**Fig 4. Phase (low frequency domain) during the phases of LTX.** Boxplot of the phase in the low frequency domain showing no differences during the preparation, anhepatic and reperfusion phase during LTX.

investigated [22]. In the latter study, patients with acute, acute-on-chronic and chronic liver failure were examined, providing a representative patient population. Interestingly, the authors found an association between impaired dCA and higher MELD score.

## Physiological data

While there are statistically significant differences in physiological data (MAP, MCBv, CI, HR, and $p_aCO_2$), these variations are clinically of minor relevance and expected within the different phases during LTX. However, the typically decrease in MAP, CI, HR, and the increase in $p_aCO_2$ in the reperfusion phase compared to the other phases of surgery may potentially cause the disturbed integrity of dCA, evaluated in the gain of the very low and low frequency domain.

## Primary outcome

The results of this observational pilot study provide a subtle indication of a more intact dCA during the preparation phase. Ardizzone et al. proposed the hypothesis, that the use of TCD and measurement of dCA could predict the function of the newly transplanted liver shortly

after reperfusion [1]. They investigated six patients, three of whom suffered liver failure due to acute hepatitis infection, two due to Amanita phalloides poisoning, and one due to isoniazid intoxication. dCA was assessed intermittently using a linear regression analysis between mean ABP and parallel changes in CBv using transcranial Doppler ultrasound during a slow phenylephrine infusion [1]. They compared dCA of the first hour (paleohepatic) to the last hour of surgery (neohepatic), while dCA recovered quickly after transplantation and was already normal towards the end of the surgery. According to the colleagues, the examination of the dCA with the help of Doppler monitoring enables the prediction of the function of the newly transplanted liver shortly after reperfusion [1]. Our data generated of 20 patients did not confirm this. Although it is not explicitly mentioned, it can be assumed that all six patients suffered acute liver failure [1]. Moreover, with an average age of 41.5 years, the patient population was considerably younger and possibly had fewer comorbidities. In our patient population, only 5% of the patients had acute liver failure, and perhaps the various mechanisms of chronic liver diseases can cause a recovery of dCA over time despite the good function of the newly implanted organ.

In addition to a connection between liver function and impaired dCA [1], portal hypertension with its systemic consequences due to advanced cirrhosis also seems to play a role. It is well known that with the progression of cirrhosis, the liver can no longer fulfill its functions such as synthesis, detoxification, storing and metabolism. Cirrhosis leads to cellular remodeling with increased vascular resistance in the liver and portal hypertension [23]. The systemic consequences are reflected in the hyperdynamic state, characterized by high cardiac output and low systemic resistance [24], which is also evident in our data. The reasons are mainly an expansion of circulating plasma volume (due to water and sodium retention) and an increase in the vasodilator NO [25]. These increased levels of circulating NO could worsen dCA by affecting the stability of neurovascular coupling, resulting in continuous cerebral vasodilation [26, 27]. In addition, patients with advanced liver cirrhosis often show a lower MAP than healthy individuals [24, 28]. Such hypotension is often well tolerated as long as end-organ perfusion is ensured in the compensated state [24, 28]. In particular, the anhepatic phase and reperfusion are hemodynamically critical phases that can throw the barely compensated state out of balance. In the anhepatic phase, depending on the surgical technique, there is partial to complete clamping of the vena cava with an abrupt influence on cardiac preload and maintenance of MAP. Even brief hypotensive episodes can lead to a deterioration in cerebral perfusion if dCA is impaired [24]. As the colleagues around L'Écuyer reveal in their review article, the pathophysiological mechanisms that lead to a deterioration of dCA in advanced liver cirrhosis are manifold. In addition to an altered neurovascular coupling, an increased GABAergic tone also seems to lead to permanent cerebral vasodilation [24]. Additionally, cirrhosis could also lead to altered endothelial cell communication due to deficient tight junctions [24]. Since the brain is very sensitive to blood-derived toxins, which could be present in greater quantities in the anhepatic phase due to the lack of clearance, an altered blood brain barrier integrity could be an explanation for an impaired dCA during the anhepatic phase. Furthermore, the analysis of the gain shift of our data exhibited a more impaired dCA in the reperfusion phase compared to the preparation phase in the very low and low frequency band. No differences were observed in the analysis of the phase shift. This subtle indication of an impaired dCA could also be attributed to the decrease of the vital parameter values, such as e.g. MAP, CI, and HR, in the reperfusion phase as well as to a potential reperfusion injury. This happens when the delivery of oxygen and nutrients to an organ is restored, leading to oxidative stress and inflammation, and is often accompanied by hypotension [29]. All these mechanisms have developed over time and have led to the changes and characteristics mentioned above in patients with liver failure. It is quite likely that these mechanisms do not regress immediately.

This could explain why dCA is still disturbed during reperfusion even in a well-functioning liver.

## Secondary outcomes

We were unable to draw closer conclusions about the relationship between the entity of liver failure and the integrity of dCA. Additionally, the underlying diseases are highly heterogeneous, with varying pathophysiologic mechanisms (e.g., autoimmune components).

Interestingly, up to 30% of patients may experience neurological complications after LTX, even without a history of HE [30]. In our study, 20% of patients developed delirium within 4 days The results in the frequency domain suggest a more intact dCA in patients with delirium. However, due to the small sample size, no valid conclusions can be drawn. While the incidence of delirium in the general population is up to 2% postoperatively [31], it increases to 10–27% in patients after LTX [31, 32]. Therefore, it can be assumed that other factors may influence cognitive functions before and after LTX.

Moreover, the mortality within 90 days after LTX is reported to be 5–15% [33, 34], and within one year, it is 10% [33]. In our study, two patients died after 41 and 54 days, respectively, and the third after 94 days. A connection to impaired dCA could not be demonstrated, in particular due to the small sample size, and the data have to be interpreted descriptively.

Several studies suggest a link between liver function and impaired dCA [1, 2, 35], but it appears to be an interaction of multiple factors [24]. Therefore, it is not surprising that no correlation was found between the laboratory data (MELD score, bilirubin level, Quick, aspartate aminotransferase, alanine aminotransferase, and glutamate dehydrogenase levels) and impairment of dCA in the present study. Although we could not demonstrate an improvement of dCA towards the end of surgery, other studies have reported an intraoperative improvement [1] and restoration of dCA after LTX within 48 hours post-surgery [35].

## Limitations and strengths

A limitation of the pilot study is that the sample size is too small relative to the heterogeneity of the underlying liver disease, and therefore also for further sub-analyses challenging. Consequently, examination of a sufficient number of patients for each disease entity and other variables is essential. Additionally, measuring dCA during the stay at the intensive care unit would have been useful to gain more insight into when and to what extent dCA normalizes postoperatively and whether there are any (in)dependent neurological complications.

One strength of the study is that it includes the highest number of patients compared to other studies on this topic. The ability to collect and exclude a relatively large amount of data on possible confounders is noteworthy. Continuous, rather than intermittent, measurement of dCA provides more insight into the dynamics of dCA [1], especially in the context of hemodynamically eventful LTX. Furthermore, the study successfully collected and assessed the one-year mortality of all included patients, an aspect not previously investigated. However, there is currently no literature addressing the pathophysiological background, offering opportunities for new investigations.

## Conclusion

In this observational pilot study, the obtained results provide a very subtle indication of a more intact dCA during the preparation phase. However, no associations were observed with underlying primary liver disease, the occurrence of delirium, the length of time spent in intensive care or hospital. Furthermore, the intactness of dCA did not demonstrate any correlation with one-year mortality. Based on these study results, it is challenging to extrapolate any

recommendations for interpreting the state of dCA during LTX and reliably identifying patients at risk.

## Supporting information

**S1 Table. Clinical data of all 20 Patients undergoing liver transplantation.**
(DOCX)

**S1 File. Cerebrovascular autoregulation during liver transplantation–raw data.**
(PDF)

## Acknowledgments

This paper contains parts of the professorial dissertation (Habilitation) of Bente Marei Wolpert.

## Author Contributions

**Conceptualization:** Patrick Schramm, Eva-Verena Griemert.

**Data curation:** Bente Marei Wolpert, David Jonas Rothgerber, Ann Kristin Rosner, Malte Brunier, Patrick Schramm, Eva-Verena Griemert.

**Formal analysis:** Robert Kuchen, Patrick Schramm.

**Investigation:** Bente Marei Wolpert, David Jonas Rothgerber, Ann Kristin Rosner, Malte Brunier, Eva-Verena Griemert.

**Supervision:** Patrick Schramm, Eva-Verena Griemert.

**Writing – original draft:** Bente Marei Wolpert, Patrick Schramm, Eva-Verena Griemert.

**Writing – review & editing:** Bente Marei Wolpert, David Jonas Rothgerber, Ann Kristin Rosner, Malte Brunier, Robert Kuchen, Patrick Schramm, Eva-Verena Griemert.

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
