## [Decision Letter · Decision Letter 0]

10 Oct 2023

PONE-D-23-26201Evaluation of cerebrovascular autoregulation during liver transplantationPLOS ONE

Dear Dr. Griemert,

Thank you for submitting your manuscript to PLOS ONE. After careful consideration, we feel that it has merit but does not fully meet PLOS ONE’s publication criteria as it currently stands. Therefore, we invite you to submit a revised version of the manuscript that addresses the points raised during the review process.

I have read with great interest the manuscript entitled “Evaluation of cerebrovascular autoregulation during liver transplantation” submitted to PLOS One. First, I would like to congratulate the authors on investigating this significant topic in liver transplantation. In addition, the manuscript is well-written and adds relevant information to the literature.

Nevertheless, the expert reviewers raised concerns about the methodological details of the study that must be addressed. Reviewer 1 has extensively reviewed the reliability and validity of the Mx and corresponding indices and concluded that it cannot be dichotomized and should not be used as a measure for cerebral autoregulation. Reviewer 2 has highlighted several aspects of the manuscript which deserve attention. Mainly, methodological details must be verified. Please see the comments and revise the manuscript carefully.

We look forward to receiving your revised manuscript.

Kind regards,

Yuri Boteon, M.D., Ph.D.

Academic Editor

PLOS ONE

2. Please ensure that you refer to Figure 2 in your text as, if accepted, production will need this reference to link the reader to the figure.

Reviewers' comments:

Reviewer's Responses to Questions

**Comments to the Author**

1. Is the manuscript technically sound, and do the data support the conclusions?

Reviewer #1: No

Reviewer #2: Yes

2. Has the statistical analysis been performed appropriately and rigorously? 

Reviewer #1: Yes

Reviewer #2: No

3. Have the authors made all data underlying the findings in their manuscript fully available?

Reviewer #1: No

Reviewer #2: No

4. Is the manuscript presented in an intelligible fashion and written in standard English?

Reviewer #1: Yes

Reviewer #2: Yes

5. Review Comments to the Author

Reviewer #1: Dear Editor and authors,

I would like to thank you for the opportunity to review this manuscript. Investigation of cerebral autoregulation during Liver transplantation is an interesting theme. The manuscript in itself is relatively well written with an easy to follow introduction, methods, results, and discussion section. However, I have some issues and comments which must be addressed.

I have spent a lot of time investigating the reliability and validity of Mx and the corresponding indices. Based on a systematic review (https://pubmed.ncbi.nlm.nih.gov/34617816/), we concluded that the methodology on how to calculate Mx varies throughout the literature, which limits comparisons between studies. This is exemplified in this observational study (https://pubmed.ncbi.nlm.nih.gov/34173717/) where different block and epoch lengths are investigated. In this study other factors seems to influence the reliability of Mx. Especially the idea of dichotomizing Mx seems flawed, since multiple studies showed that Healthy volunteers on average have an Mx of above 0.3 or even 0.45 which is recommended by Schmidt et al. [1–3]. Furthermore, the validity is questionable (https://pubmed.ncbi.nlm.nih.gov/35343649/;
https://pubmed.ncbi.nlm.nih.gov/36008917/). Thus, I do not believe Mx should be used to assess cerebral autoregulation. Preliminary results (unfortunately unpublished) provide somewhat better reliability for Transfer Function Analysis. We have developed an R-package which can calculate TFA (https://cran.r-project.org/web/packages/clintools/index.html). To get started with TFA I would suggest looking into https://pubmed.ncbi.nlm.nih.gov/26782760/ and https://pubmed.ncbi.nlm.nih.gov/35962478/. Furthermore Jürgen Claassen have provided a great overview for the current knowledge of cerebral autoregulation which is rather long, but for some very interesting (https://pubmed.ncbi.nlm.nih.gov/33769101/). TFA has its own issues, but I do not have any data concluding that TFA should not be used as a measure of cerebral autoregulation.

In conclusion, I do not believe Mx can be dichotomized and it should not be used as a measure for cerebral autoregulation.

References:

1. Ortega-Gutierrez S, Petersen N, Masurkar A, Reccius A, Huang A, Li M, et al. Reliability, asymmetry, and age influence on dynamic cerebral autoregulation measured by spontaneous fluctuations of blood pressure and cerebral blood flow velocities in healthy individuals. J Neuroimaging. 2013/04/24. 2014;24:379–86.

2. Reinhard M, Wehrle-Wieland E, Roth M, Niesen WD, Timmer J, Weiller C, et al. Preserved dynamic cerebral autoregulation in the middle cerebral artery among persons with migraine. Exp Brain Res. 2007/02/07. 2007;180:517–23.

3. Yam AT, Lang EW, Lagopoulos J, Yip K, Griffith J, Mudaliar Y, et al. Cerebral autoregulation and ageing. J Clin Neurosci. 2005;12:643–6.

Reviewer #2: Authors present a cerebral autoregulation analysis in patients undergoing liver transplant. Data are acquired in the different phases of surgery.

The paper is interesting and original. However scanty data and analyses are presented and the work should be improved in this sense.

Introduction: there is lack of reference to some previous works analyzing correlation of MAP and MCBv in surgery. The use of correlation instead of other index of cerebral autoregulation (E.g. Autoregulation Index: Tiecks et al., Stroke. 1995 Jun;26(6):1014-9. doi: 10.1161/01.str.26.6.1014.; Gelpi et al., Auton Neurosci. 2022 Jan;237:102920. doi: 10.1016/j.autneu.2021.102920. or Transfer function analysis Panerai et al.,J Cereb Blood Flow Metab. 2023 Jan;43(1):3-25. doi: 10.1177/0271678X221119760..) should be better motivated.

Methods:

How did you syncronize the CBFV and AP signals? If signals were not properly synchronized the correlation could differ.

For consistency with the other works evaluating CA I would suggest you to use "CBv" instead of "CBFV" and "MCBv" in place of "MCBFV". (See the White paper on CA update, Panerai et al., J Cereb Blood Flow Metab. 2023 Jan;43(1):3-25. doi: 10.1177/0271678X221119760.)

Why did you use 6-seconds windows average on the signals and not provide first a beat-to-beat estimation of the mean CBv and mean AP (MAP)?

A reference would be needed.

How long were the recordings? How many windows were calculated and how did authors obtained a single value for acquisition phase?

Please clarify and consider to use a figure to explain the computations.

Line 139: define the MELD score and how it is calculated.

A table resuming demographic and clinical characteristics in terms of average value in the acquired population would improve the work.

Paper evaluating CA normally evaluate it from series of mean arterial pressure (MAP) and mean cerebral blood velocity (MCBv) extracted on a beat-to-beat basis or in opportune windows.

That should be done for the three different experimental phases and the classical markers of mean and variance (or similar) of MAP and MCBv should be given.

Furthermore, the statistical analysis is not completely clear. Which conditions were compared with a t-test?

Why not performing an ANOVA among the three phases of intervention to observe if in the overall population there were differences in the experimental phases?

The significance of the Pearson’s correlation coefficient should be given. Anyway, the significance of the coupling should be tested via a surrogate analysis to check if the real value of correlation is different from a random coupling (see the one proposed in Bari et al. BPSC 2021, https://doi.org/10.1016/j.bspc.2021.102735, in similar conditions, or use e.g. an Iteratively-refined amplitude adjusted Fourier transform-based method(Schreiber 1996, Phys. Rev. Lett. 77 635–8 DOI:10.1103/PhysRevLett.77.635)

Please pay atention that Raw data, e.g. time series or recordings are not declared as fully available as requested by the journal.

6. PLOS authors have the option to publish the peer review history of their article (what does this mean?). If published, this will include your full peer review and any attached files.

Reviewer #1: **Yes: **Markus Harboe Olsen

Reviewer #2: No

---

## [Author Response · Author response to Decision Letter 0]

24 Jan 2024

Dear Prof. Boteon, dear reviewers, 

we are pleased to resubmit our comprehensively revised manuscript.

Our detailed comments are provided in the response-to-reviewer letter.

With kind regards

Eva-Verena Griemert

---

## [Decision Letter · Decision Letter 1]

11 Mar 2024

PONE-D-23-26201R1Evaluation of dynamic cerebrovascular autoregulation during liver transplantationPLOS ONE

Dear Dr. Griemert,

Thank you for submitting your manuscript to PLOS ONE. After careful consideration, we feel that it has merit but does not fully meet PLOS ONE’s publication criteria as it currently stands. Therefore, we invite you to submit a revised version of the manuscript that addresses the points raised during the review process.

The expert reviewers agree that the manuscript has improved significantly. Nevertheless, Reviewer 1 has provided information regarding their experience recommending not using Mx and derived indices. Please read the reference provided and elaborate on the topic. Reviewer 2 suggested amendments to the figures and captions. Please amend the manuscript accordingly.

We look forward to receiving your revised manuscript.

Kind regards,

Yuri Boteon, M.D., Ph.D.

Academic Editor

PLOS ONE

Journal Requirements:

Reviewers' comments:

Reviewer's Responses to Questions

**Comments to the Author**

1. If the authors have adequately addressed your comments raised in a previous round of review and you feel that this manuscript is now acceptable for publication, you may indicate that here to bypass the “Comments to the Author” section, enter your conflict of interest statement in the “Confidential to Editor” section, and submit your "Accept" recommendation.

Reviewer #1: (No Response)

Reviewer #2: All comments have been addressed

2. Is the manuscript technically sound, and do the data support the conclusions?

Reviewer #1: Partly

Reviewer #2: Yes

3. Has the statistical analysis been performed appropriately and rigorously? 

Reviewer #1: N/A

Reviewer #2: Yes

4. Have the authors made all data underlying the findings in their manuscript fully available?

Reviewer #1: No

Reviewer #2: No

5. Is the manuscript presented in an intelligible fashion and written in standard English?

Reviewer #1: Yes

Reviewer #2: Yes

6. Review Comments to the Author

Reviewer #1: Dear authors

I would like to thank the reviewer for improving the manuscript. Unfortunately, we have recently published an article discussing why Mx and the derived indicies (Mxc, Mxa and nMxa) should not be used (https://physoc.onlinelibrary.wiley.com/doi/10.1113/EP091327). I suggest removing Mx from the manuscript and focus on TFA.

Reviewer #2: The paper has been improved and the questions raised by the reviewers have been addressed.

I would suggest authors to futher improve the clarity of figures better describing in figure captions the meaning of the symbols used for statistical significance (e.g. * means p<0.05 reperfusion vs preparation phase), or using a line between boxplots resulting significantly different instead of a single symbol.

7. PLOS authors have the option to publish the peer review history of their article (what does this mean?). If published, this will include your full peer review and any attached files.

Reviewer #1: **Yes: **Markus Harboe Olsen

Reviewer #2: No

---

## [Author Response · Author response to Decision Letter 1]

25 Apr 2024

Dear Editor Prof. Boteon,

dear Referees,

thank you very much for your second review of the revised manuscript according to your first given expert questions and suggestions. We thoroughly studied your second assessments and have revised the manuscript accordingly. Furhter details are given in the "Response to Reviewer" lertter. We hope, we answered your expectations and thank you very much for the efforts to improve the manuscript.

Yours sincerely

Eva-Verena Griemert

---

## [Decision Letter · Decision Letter 2]

4 Jun 2024

Evaluation of dynamic cerebrovascular autoregulation during liver transplantation

PONE-D-23-26201R2

Dear Dr. Griemert,

We’re pleased to inform you that your manuscript has been judged scientifically suitable for publication and will be formally accepted for publication once it meets all outstanding technical requirements.

Kind regards,

Yuri Longatto Boteon, M.D., Ph.D.

Academic Editor

PLOS ONE

Additional Editor Comments (optional):

All the comments have been appropriately addressed. The manuscript has significantly improved and is now suitable for publication.

Reviewers' comments:

Reviewer's Responses to Questions

**Comments to the Author**

1. If the authors have adequately addressed your comments raised in a previous round of review and you feel that this manuscript is now acceptable for publication, you may indicate that here to bypass the “Comments to the Author” section, enter your conflict of interest statement in the “Confidential to Editor” section, and submit your "Accept" recommendation.

Reviewer #1: (No Response)

Reviewer #2: All comments have been addressed

2. Is the manuscript technically sound, and do the data support the conclusions?

Reviewer #1: (No Response)

Reviewer #2: Yes

3. Has the statistical analysis been performed appropriately and rigorously? 

Reviewer #1: (No Response)

Reviewer #2: Yes

4. Have the authors made all data underlying the findings in their manuscript fully available?

Reviewer #1: (No Response)

Reviewer #2: No

5. Is the manuscript presented in an intelligible fashion and written in standard English?

Reviewer #1: (No Response)

Reviewer #2: Yes

6. Review Comments to the Author

Reviewer #1: (No Response)

Reviewer #2: All the comments have been addressed, the paper has been improved and is now suitable for publication.

7. PLOS authors have the option to publish the peer review history of their article (what does this mean?). If published, this will include your full peer review and any attached files.

Reviewer #1: No

Reviewer #2: No

---

## [Editor Report · Acceptance letter]

18 Jun 2024

PONE-D-23-26201R2 

PLOS ONE

Dear Dr. Griemert, 

I'm pleased to inform you that your manuscript has been deemed suitable for publication in PLOS ONE. Congratulations! Your manuscript is now being handed over to our production team.

Kind regards, 

on behalf of

Prof. Yuri Longatto Boteon 

Academic Editor

PLOS ONE